# Auxin Dynamics and Transcriptome–Metabolome Integration Determine Graft Compatibility in Litchi (*Litchi chinensis* Sonn.)

**DOI:** 10.3390/ijms26094231

**Published:** 2025-04-29

**Authors:** Zhe Chen, Tingting Yan, Mingchao Yang, Xianghe Wang, Biao Lai, Guolu He, Farhat Abbas, Fuchu Hu

**Affiliations:** 1Institute of Tropical Fruit Trees, Hainan Academy of Agricultural Sciences/Key Laboratory of Genetic Resources Evaluation and Utilization of Tropical Fruits and Vegetables (Co-construction by Ministry and Province), Ministry of Agriculture and Rural Affairs/Key Laboratory of Tropical Fruit Tree Biology of Hainan Province, Haikou 571100, China; chenzhe@hnaas.org.cn (Z.C.); yantt_hnaas@163.com (T.Y.); yangmc_hnaas@163.com (M.Y.); wangxh198@126.com (X.W.); 2Sanya Research Institute, Hainan Academy of Agricultural Sciences, Sanya 572025, China; 3School of Advanced Agriculture and Bioengineering, Yangtze Normal University, Chongqing 408100, China; laibiaoscau@163.com (B.L.); guoluhe@163.com (G.H.)

**Keywords:** *Litchi chinensis* Sonn., grafting, secondary metabolites, phytohormones, auxin

## Abstract

Grafting is a prevalent horticultural technique that enhances crop yields and stress resilience; nevertheless, compatibility issues frequently constrain its efficacy. This research examined the physiological, hormonal, and transcriptional factors regulating compatibility between the litchi (*Litchi chinensis* Sonn.) cultivars Feizixiao (FZX) and Ziniangxi (ZNX). The anatomical and growth investigations demonstrated significant disparities between compatible (FZX as scion and ZNX as rootstock) and incompatible (ZNX as scion and FZX as rootstock) grafts, with the latter showing reduced levels of indole acetic acid (IAA). Exogenous 1-naphthalene acetic acid (NAA) application markedly improved the graft survival, shoot development, and hormonal synergy, whereas the auxin inhibitor tri-iodobenzoic acid (TIBA) diminished these parameters. The incompatible grafts showed downregulation of auxin transporter genes, including ATP-binding cassette (ABC) transporter, AUXIN1/LIKE AUX1 (AUX/LAX), and *PIN*-*FORMED* (*PIN*) genes, suggesting impaired vascular tissue growth. Metabolomic profiling revealed dynamic interactions between auxin, salicylic acid, and jasmonic acid, with NAA-treated grafts exhibiting enhanced levels of stress-responsive metabolites. Transcriptome sequencing identified differentially expressed genes (DEGs) linked to auxin signaling (ARF, GH3), seven additional phytohormones, secondary metabolism (terpenoids, anthocyanins, and phenylpropanoids), and ABC transporters. Gene ontology and KEGG analyses highlighted the significance of hormone interactions and the biosynthesis of secondary metabolites in successful grafting. qRT-PCR validation substantiated the veracity of the transcriptome data, emphasizing the significance of auxin transport and signaling in effective graft development. This study provides an in-depth review of the molecular and physiological factors influencing litchi grafting. These findings provide critical insights for enhancing graft success rates in agricultural operations via targeted hormonal and genetic approaches.

## 1. Introduction

Humans have employed grafting for millennia [1]. Grafting denotes the amalgamation of plant components to develop vascular harmony, leading to an integrated plant that operates as a singular plant entity [2]. The top branch section of a single plant, termed the ‘scion’, is typically implanted onto the lower section of the other plant, referred to as the ‘rootstock’ (Figure 1). The chimera, composed of the scion and rootstock, persists as a distinct entity following the processes of wound closure. Natural grafting, which takes place once the stems or roots of plants connect and merge, has enabled the development of traditional grafting methods [3]. Recently, a significant portion of commercial fruit cultivation, along with that of certain vegetables, has depended on grafting techniques utilizing rootstocks. This practice enhances the scion’s resilience to soil-borne pathogens and environmental stresses while also modifying its vigor and yield [4]. Grafting can influence resistance to environmental stressors, and trials with cherry tomatoes demonstrated that grafting onto drought-resistant rootstocks increased fruit yield [5]. Numerous studies have indicated that grafting onto Cucurbita rootstocks can mitigate salt stress in cucumber, enhance citrus tolerance to boron stress [6], and improve tomato resistance to thermal stress [7], hence offering additional avenues for investigating the mechanisms behind these effects.

Grafting is the most expedient technique for the extensive vegetative growth of preferred fruit plants [8]. Certain economic fruiting plants are challenging to reproduce using alternative means, such as cuttings or air layering, but they exhibit a favorable response to grafting [9]. Furthermore, several varieties exhibiting exceptional fruit traits have inadequate root systems or are prone to nematodes or diseases; thus, the scion strength could be enhanced through grafting [10]. Currently, grafting is utilized for the commercial multiplication of numerous fruit trees, notably mangoes, apricots, apples, persimmons, plums, grapes, citrus, peaches, pears, and sweet cherries, characterized by high heterozygosity and difficulty in rooting from cuttings. The capacity of specific fruit tree rootstocks to diminish the size of their scions, recognized for centuries, is utilized in agriculture. Fruit trees are grafted to facilitate propagation, bypassing the juvenile phase, altering plant growth, or enhancing stress resistance, as observed in the litchi industry. The compatibility (i.e., the ability of a graft to grow effectively) between the rootstock and the scion is just as important as technical and phytosanitary regulations regarding the effectiveness of grafting. Despite its significant advantages, the comprehension of the physiological, biochemical, and molecular foundations of grafting in fruit trees remains limited; such comprehension might greatly assist breeding and commercialization endeavors.

The formation of graft unions, beginning with the initial wounding response and progressing through alterations in the cell wall, culminates in establishing vascular associations between the scion and rootstock. The process of graft union formation in plants necessitates significant restructuring of gene regulation, the translation of proteins, and metabolic pathways [8,11]. Plant hormones have particular functions in forming a functioning graft union, including mediating pectin release to promote tissue adherence, facilitating the establishment of de-differentiated callus cells, developing cellular junctions (plasmodesmata), and initiating cell division in the cambium, cortex, and pith cells adjacent to the phloem and xylem [12]. Plant hormones such as auxin, ethylene (ET), cytokinin (CK), gibberellin (GA), abscisic acid (ABA), and jasmonic acid (JA) increasingly regulate various vital cellular processes at the graft junction. However, the roles of phytohormones and other factors influencing the grafting mechanism remain largely unclear.

Litchi (*Litchi chinensis* Sonn.) is among the most popular fruit trees worldwide, particularly in tropical and subtropical regions. Grafting techniques have been utilized in its production in China for several centuries. Nonetheless, the efficacy of grafting is inconsistent due to incompatibilities between rootstocks and scions. Research on grafting compatibility and incompatibility is limited, and the metabolic and molecular pathways involved in grafting in litchi remain poorly known. In this study, we applied 1-naphthalene acetic acid (NAA) and tri-iodobenzoic acid (TIBA) during litchi grafting and examined their impact on the grafting survival rate, leaf count, shoot length and diameter, and internode length. We investigated the effects of auxin treatment on the expression of auxin transport genes and the genes involved in hormone production and signaling during grafting in litchi. We focused on the basic mechanisms of graft union formation in litchi, especially the impact of auxin on the entire process, while previous studies mainly examined the interaction network between the rootstock and scion, specifically how compatibility between the two affects the growth, flowering, fruiting, and fruit yield and quality of grafted varieties. We investigated the likely mechanisms via transcriptome and hormone metabolome analysis, providing an invaluable resource for clarifying the hormonal impact on rootstock and scion effectiveness in diverse biological mechanisms.

## 2. Results

### 2.1. Physiological and Anatomical Observation of Litchi Grafts

We employed the cultivars Feizixiao (FZX) and Ziniangxi (ZNX) for grafting to investigate the mechanisms of compatibility and incompatibility in litchi. Figure 1A,B illustrate the growth performance of litchi grafts in field conditions, emphasizing the distinctions between compatible and incompatible grafts across different post-grafting intervals. The images reflect the health and growth patterns of the grafts, which were systematically monitored during the study. Our findings indicate that ‘FZX’ is compatible as a scion with ‘ZNX’ as the rootstock, whereas the reverse combination is incompatible (Figure 1C,D). The contact strengthened with time as the cells interdigitated and connected the rootstock and scion together. The stem cell-like tissue differentiated and gave rise to new vascular tissues, which connected the xylem and phloem between the scion and rootstock (Figure 1E). The rootstock and scion were not in close contact, and there was a large gap between them. The vascular complexes were abnormal (Figure 1F), which offers insights into the biological mechanisms influencing graft success. Auxin is a crucial hormone that determines the compatibility and incompatibility of litchi grafts; thus, we measured the total indole acetic acid (IAA) content in both the compatible and incompatible grafts at multiple time points (Figure 1G). The results showed significant variations in IAA contents between the graft types, with the incompatible grafts exhibiting notably lower IAA levels. The data suggest that IAA plays a crucial role in the success of grafting, and the differences observed may contribute to the failure or success of graft unions. A gene expression analysis (Figure 1H) focused on ABC transporter, Aux/LAX, and PIN genes, showing distinct patterns between compatible and incompatible grafts at 7, 14, and 21 days post-grafting. These differences in gene expression highlight the molecular mechanisms that might influence graft compatibility. These observations suggest the potential biological factors contributing to the success or failure of grafting.

### 2.2. Effect of NAA and TIBA Treatments on Graft Performance

To gain a better understanding of the elements that influence the grafting process, we investigated compatible litchi grafts with exogenous NAA and its inhibitor TIBA (Figure 2A). The findings indicated that exogenous NAA significantly improved the graft survival, whereas TIBA reduced it (Figure 2B). The plant survival rate was improved by 13% post-NAA application compared to the control, while the TIBA application substantially decreased it (nearly 24%). Furthermore, the number of leaves, shoot length, shoot diameter, and internode length were enhanced in grafts treated with exogenous NAA compared to the control and TIBA-treated litchi grafts (Figure 2C–F), implying that auxin plays an integral role in the effective grafting process of litchi.

### 2.3. Hormonal Metabolic Profiling

To enhance our understanding of the effectiveness of exogenous NAA application on the compatible litchi grafting mechanism, we utilized UPLC–MS/MS-based targeted metabolomics to analyze plant hormones following NAA and TIBA application, as compared to the control group. In Figure 3, the results of the metabolomic analysis of the hormonal content reveal significant differences in the hormone levels between the control, NAA-treated, and TIBA-treated grafts. The auxin and salicylic acid levels were higher than those of other phytohormones (Figure 3A). Likewise, the concentrations of auxin and salicylic acid were highest relative to those of other phytohormones (Figure 3B). The concentration of salicylic acid surged by 71% two hours (CK1) and seven days (CK3) post-grafting, relative to that for the control (61%). In contrast, the auxin concentration was markedly reduced from 30% (NAA1) to 20% (control). Notably, the ABA concentration was significantly elevated three days post-grafting with TIBA application (8%) in comparison to the control or NAA application. The heatmap illustrates the hormone metabolites exhibiting unique expression patterns at various time intervals (Figure 3C). The majority of the metabolites were auxins, whereas melatonin-related metabolites were the least prevalent. The auxin levels significantly increased three days after grafting following NAA administration, whereas TIBA produced the opposite effect. In short, jasmonic acid and salicylic acid were prevalent during grafting after NAA treatment. A K-means clustering analysis categorized them into six distinct groups according to their abundance at various stages (Appendix A). The Venn diagrams (Figure 3D,E) also display differentially expressed hormones, revealing complex interactions between the NAA and TIBA treatments. Totals of 12, 7, 8, and 7 phytohormones were identified in the comparisons of CK1 with NAA1, CK2 with NAA2, CK3 with NAA3, and CK4 with NAA4, respectively (Figure 3D). Similarly, 10, 9, 5, and 12 phytohormones were identified in the comparisons of CK1 with TIBA1, CK2 with TIBA2, CK3 with TIBA3, and CK4 with TIBA4, respectively (Figure 3E).

### 2.4. Transcriptome Analysis

An RNA-seq analysis was conducted on compatible litchi grafts at 2 h, 3 days, 7 days, and 14 days post-grafting to identify crucial genes affecting the grafting mechanism. Figure 4A,B demonstrate significant differences among the groups, but there were no differences within each group’s samples, indicating high data accuracy based on the principal component analysis (PCA) and heatmap analysis. The transcriptome dataset produced 6.87–10.01 GB of clean bases, with an error rate below 0.01%, a Q20 value exceeding 97.6%, a Q30 value surpassing 93.24%, and a GC content ranging from 43.67% to 46.58% (Appendix A). The correlation analysis revealed high homogeneity within samples (Appendix A). The quantity of downregulated DEGs exceeded that of upregulated DEGs (Figure 4C). In the comparisons of CK1 vs. NAA1, CK2 vs. NAA2, CK3 vs. NAA3, and CK4 vs. NAA4, 84, 38, 126, and 39 DEGs were upregulated, while 82, 65, 295, and 59 were downregulated, respectively. In the comparisons of CK1 vs. TIBA1, CK2 vs. TIBA2, CK3 vs. TIBA3, and CK4 vs. TIBAA4, 26, 61, 6, and 47 DEGs were upregulated, while 23, 81, 7, and 473 were downregulated, respectively.

The Upset R plot highlighted the common and specifically expressed DEGs among the samples (Figure 4D,E). Among the elevated DEGs, 3 genes were consistently detected across all samples, whereas 108, 66, 47, 38, 34, 25, 8, and 3 were uniquely expressed in the comparisons of CK1 vs. NAA1, CK2 vs. NAA2, CK3 vs. NAA3, CK4 vs. NAA4, CK1 vs. TIBA1, CK2 vs. TIBA2, CK3 vs. TIBA3, and CK4 vs. TIBA4, respectively (Figure 4D). Among the downregulated DEGs, 3 genes were consistently identified across all samples, while the unique expressions numbered 430, 276, 52, 52, 46, 23, 8, and 2 for the comparisons of CK1 vs. NAA1, CK2 vs. NAA2, CK3 vs. NAA3, CK4 vs. NAA4, CK1 vs. TIBA1, CK2 vs. TIBA2, CK3 vs. TIBA3, and CK4 vs. TIBA4, respectively (Figure 4E). A K-means cluster analysis further classified all the DEGs into two subclasses (Figure 4F). Subclass 1 and subclass 2 contained 6416 and 7303 DEGs. Subclass 1 DEGs had the highest expression levels in CK2, CK4, NAA2, NAA4, TIBA2, and TIBA4, whilst subclass 2 DEGs were primarily expressed in CK1, CK3, NAA1, NAA3, TIBA1, and TIBA3.

### 2.5. GO Enrichment and KEGG Pathway Analyses

The GO assessment revealed that the majority of DEGs were associated with the following GO terms, which were prevalent after the application of NAA and TIBA post-grafting: secondary metabolite biosynthetic process (GO:0044550), terpene synthase activity (GO:0010333), phenylpropanoid biosynthetic process (GO:0009699), cellular response to decreased oxygen levels (GO:0036294), carbon–oxygen lyase activity, acting on phosphates (GO:0016838), transmembrane receptor protein kinase activity (GO:0019199), and terpene biosynthetic process (GO:0046246) (Appendix A). The KEGG pathway analysis indicated that the majority of the DEGs were significantly linked to the pathways of ‘plant hormone signal transduction’, ‘MAPK signaling pathway’, ‘biosynthesis of secondary metabolites’, ‘monoterpenoid biosynthesis’, and ‘flavonoid and phenylpropanoid biosynthesis’. The quantity of DEGs linked to GO terms was predominantly downregulated (Figure 5A). A total of 5928 DEGs were assigned to gene ontology (GO) concepts; of these, 4089 were downregulated, and 1803 were upregulated. The quantity of DEGs was greater in the comparisons of CK4 with TIBA4 (1736 downregulated and 178 upregulated) and CK3 versus NAA3 (1239 downregulated and 566 upregulated). A comparable trend was noted among the DEGs associated with the KEGG pathways, wherein the majority of DEGs were downregulated, especially after the TIBA application (Figure 5B). Nevertheless, the number of differentially expressed genes was higher 7 days post-NAA (CK3 vs. NAA3) and 14 days post-TIBA (CK4 vs. TIBA4) application.

### 2.6. Differential Expression Pattern of Genes Related to Secondary Metabolites and Hormonal Pathways

ATP-binding cassette (ABC) transporter genes are pivotal in the transfer of phytohormones within the plant and to the external environment [13,14]. Consequently, the expression of ABC transporter genes in the grafting mechanism after NAA and TIBA treatment was examined (Figure 5C,D). About 8 ABC transporter and 14 terpenoid biosynthesis-related genes were identified from the annotation data. The data indicated that the expression of ABC transporter genes increased following the application of NAA and TIBA (Figure 5C). A similar expression pattern of terpenoid biosynthesis-related genes was noted after phytohormone treatment, particularly two hours post-grafting (Figure 5D). Nonetheless, the expression of two terpene synthase genes (*LITCHI031167* and *LITCHI004927*) was heightened in the later phases of the grafting period, though they exhibited minimal expression during the initial stage of grafting.

Phytohormones govern all facets of plant development and reactions to biotic and abiotic stressors. The data analysis of genes annotated to plant hormone signaling transduction and secondary metabolite pathways revealed eight distinct hormone biosynthesis and signaling pathways common to all identified genes. The pathways encompassed abscisic acid, auxin, brassinosteroids, jasmonic acid, gibberellins, ethylene, salicylic acid, and cytokinins (Figure 6). Totals of 26, 11, 12, 7, 12, 13, 13, and 11 DEGs were identified concerning auxin, gibberellic acid, abscisic acid, brassinosteroids, ethylene, salicylic acid, jasmonic acid, and cytokinins, respectively. During the initial phases after grafting, the levels of most Aux/IAA and GH3 genes were elevated, but PIF4, TIR1, and particularly ARF genes exhibited significant expression levels in the later stages following treatment application. In a similar vein, the majority of ethylene genes (except LITCHI028323 and LITCHI024767), as well as jasmonic acid-related genes including jasmonate-ZIM domain (JAZ), MYC, and 12-oxo-phytodienoic acid-10,11-reductase (OPR2), exhibited significant responsiveness during the initial phase of grafting, while jasmonate-induced oxygenase 1 (JOX1), OXI1, methyl jasmonate esterase 1 (MJE1), and C-terminal domain phosphatase-like 1 (CPL1) demonstrated increased expression in later phases. Among the gibberellins, GA2ox exhibited the highest expression level during the early stage, whereas the others demonstrated peak expression during the latter phases of grafting. Abscisic acid, PHOSPHATASE 2 C (PP2C), pyrabatin resistance 1-like (PYL), and most of the Snf1-Related protein Kinase (SnRK) proteins demonstrated elevated expression during the later stages of grafting. The majority of NONEXPRESSOR OF PATHOGENESIS-RELATED GENES 4 (NPR4) displayed increased expression during the early period, whereas NPR1 and TGA1 exhibited the opposite trend among the salicylic acid-related genes. Brassinosteroid- and cytokinin-related genes demonstrated the same pattern, with the majority exhibiting peak expression during the later stages after grafting.

Among the secondary metabolites, the majority of DEGs were linked to the biosynthesis pathways of terpenoids, anthocyanin, and phenylpropanoids. We identified 51 DEGs associated with the anthocyanin and phenylpropanoid biosynthesis pathways (Figure 7). Anthocyanins are classified as flavonoids, sharing a common synthetic pathway with various flavonoids. They are produced through branching synthesis reactions originating from dihydrokaempferol. Figure 7 illustrates that the majority of these genes demonstrated increased abundance in the later phases of grafting and showed a significant response following the application of NAA and TIBA as compared to the control group. The expression levels of phenylalanine ammonia-lyase (PAL), cinnamic acid 4-hydroxylase (C4H), MaT, dihydrofl flavonol reductase (DFR), 3-rhamnosyl transferase (3RT), and isoflavone reductase (IFR) were highest during the early phases of the grafting period following the application of NAA and TIBA. The majority of these genes exhibited a high responsiveness to NAA application in comparison to the TIBA or control group.

### 2.7. Validation of Differentially Expressed Genes

Figure 8 presents the quantitative RT-PCR validation of 12 selected genes. The results confirm the differential expression of these genes, with FPKM values provided for comparison. The data further validate the RNA sequencing findings, supporting the observed trends in gene expression related to litchi graft compatibility and treatment response.

## 3. Discussion

Physiological and anatomical analyses of litchi grafting yield a fundamental understanding of the governing principles of graft compatibility and incompatibility. Our findings indicated that ‘FZX’ as a scion on ‘ZNX’ rootstock achieved efficient graft compatibility; nevertheless, the inverse combination displayed incompatibility. The anatomical evaluations confirmed these findings, revealing substantial structural variations between the compatible and incompatible transplants. These data correspond with results from prior studies indicating that effective grafting is significantly reliant on the cellular and vascular linkages between scions and rootstocks [15,16,17]. The application of auxin to the rootstock and scion enhances the grafting success rate in Chinese hickory and identifies hickory genes within the PIN, ABCB, and AUX/LAX families, which are known to encode influx and efflux carriers involved in the polar transport of auxin [18]. These findings indicated that heightened expression of various genes, including *CcPIN1b* and *CcLAX3*, correlates with effective grafting, consistent with our results.

A primary factor influencing graft success seems to be auxin, namely IAA, the level of which was observed to be markedly reduced in incompatible grafts. It is well established that auxin is crucial for grafting because of its central role in cell differentiation and the development of vascular tissue [19,20]. The study of gene expression in ABC transporter, Aux/LAX, and PIN genes corroborates this concept, revealing differing variations in expression across compatible and incompatible grafts at various post-grafting intervals. The decreased expression of auxin-related genes in incompatible grafts could impede their success by disrupting vascular tissue growth and limiting nutrient transfer [21,22,23]. By considerably increasing the survival rates and improving morphological characteristics such as the shoot length, leaf number, and internode length, the exogenous application of NAA provided more evidence that auxin plays an integral part in maintaining the viability of grafts. The use of TIBA, an auxin transport inhibitor, led to declined graft survival rates, signifying that adequate auxin transport is crucial for effective grafting. Previous research indicated that the application of NAA to scion apices enhanced vascular reconnection during the grafting of cactus species. In *Arabidopsis thaliana*, auxin accumulation in the grafted region precedes cell development and vascular reconnection between the rootstock and scion [12,19,24]. Our findings revealed that IAA levels, along with those of the majority of PIN, Aux/LAX, and ABC transporter genes, were markedly increased in compatible grafts compared to incompatible grafts, presumably associated with callus formation and improved nutrition transfer, as evidenced by anatomical images demonstrating superior transport influx. These findings align with those of previous research that showed the beneficial effects of auxin treatment on graft development and vascular connection [1,25,26].

This particular metabolomic study indicated substantial hormonal changes after the application of NAA and TIBA, with auxin and salicylic acid identified as the principal phytohormones affecting graft compatibility. The noted elevation in salicylic acid and jasmonate concentrations during grafting indicates their role in stress responses and immunological signaling, perhaps aiding in graft union formation [27,28]. The divergent impacts of NAA and TIBA on auxin concentrations underscore the essential function of auxin transport and metabolites in determining the effectiveness of grafting. The significance of JA in NAA-treated grafts indicates auxin–JA synergy in tissue regeneration, as JA is recognized for its ability to promote cell proliferation in damaged tissues [8]. These findings reflect the hormone feedback seen in tomato and apple grafts, where optimal auxin–SA–JA ratios enhanced union efficiency [26,29,30]. The increased ABA levels following TIBA treatment suggest an adaptation to stress that could potentially contribute to graft failure [31,32].

Our transcriptome research yielded profound insights into the genetic foundation that underlies graft success, identifying a substantial number of differentially expressed genes (DEGs) across various treatment settings. The transcriptomic studies identified significant differentially expressed genes associated with hormone signaling (ARF, GH3, Aux/IAA) and secondary metabolism (Figure 5, Figure 6 and Figure 7). The upregulation of ABC transporters (Figure 5C) corresponds to their function in auxin efflux and detoxification [33,34]. Additionally, the stage-specific expression of terpenoid biosynthesis genes (*LITCHI031167* and *LITCHI004927*) indicates that terpenoids may strengthen graft unions by improving membrane integrity or providing antimicrobial defense [35,36]. Significantly, a greater number of genes exhibited downregulation compared to upregulation, especially in TIBA-treated grafts, which corresponds to the noted reduction in auxin levels. The pathway analysis conducted using KEGG indicated that these DEGs were abundant in pathways associated with plant hormone signal transduction, secondary metabolite production, and MAPK signaling, all of which are crucial for successful grafting (Figure 5 and Figure 6; Appendix A). The enhancement of the phenylpropanoid and anthocyanin pathways (Figure 7) indicates lignin and flavonoid accumulation in graft interactions, which are vital processes for mechanical strength and stress-induced resilience in grapevine and citrus grafts [4,8,37,38]. The biosynthesis pathways of secondary metabolites, specifically those related to terpenoids, anthocyanins, and phenylpropanoids, demonstrated variations in gene expression in response to NAA and TIBA treatments. Increased expression of genes involved in the biosynthesis of anthocyanins and flavonoids in grafts treated with NAA raises the possibility that these metabolites have a role in graft success, maybe through boosting antioxidant activity and coping with stress [39,40]. The anthocyanin biosynthesis pathway involves a series of genes, categorized into early pathway genes (PAL, C4H, 4CL, CHS, CHI, and F3H) and late pathway genes (DFR, ANS, and UFGT). The significant suppression of MAPK and ethylene-responsive genes (e.g., ERF1) in TIBA-treated grafts (Figure 6 and Figure 7) underscores the function of auxin as a transcriptional trigger in wound-responsive pathways [41]. These findings further substantiate that auxin transport is essential for effective grafting (Figure 9). The pronounced differential expression of genes associated with hormone production, transport, and signaling indicates that these molecular mechanisms are vital factors in determining graft compatibility and effectiveness in litchi. This work elucidated the physiological, hormonal, and genetic principles underlying litchi grafting and offers potential techniques to enhance graft success rates in horticulture. Future research must be conducted to further investigate the pivotal role of auxin transport and signaling mechanisms in heterografts, as the present study was limited to compatible grafts.

## 4. Materials and Methods

### 4.1. Plant Materials and Grafting

*Litchi chinensis* Sonn. cultivars ‘Feizixiao’ (FZX) and ‘Ziniangxi’ (ZNX) were utilized in the innovative experimental orchard of the Hainan Academy of Agricultural Sciences (19°23′~20°01′ N, 109°45′~110°15′ E), Haikou, China. The rationale for selecting reciprocal grafts between ‘Feizixiao’ (FZX) and ‘Ziniangxi’ (ZNX) was based on both practical horticultural challenges and prior evidence of directional incompatibility in this cultivar pair. FZX and ZNX are dominant commercial litchi cultivars prized for their fruit quality and yield. The cultivar ‘ZNX’ was grafted onto ‘FZX’ (heterograft), representing an incompatible combination, whereas ‘FZX’ was grafted onto itself (homograft), indicating a compatible combination. Tender branches were harvested from mature litchi plants for top grafting.

### 4.2. Hormonal Treatment and Morphological Parameter Analysis

The scion and rootstock of the homograft used in the experiment were treated with 100 mg/L NAA, 100 mg/L TIBA, or water before grafting. All samples were collected from the graft interfaces of the combination 2 h, 7 days, 14 days, and 30 days after grafting, and each stage of the combination had three samples. The samples were stored immediately in liquid nitrogen and then stored at −80 °C until use.

The grafting survival rate, which is the proportion of plants that survived the grafting procedure, was assessed one month post-grafting. Upon the resumption of normal development by the scion, typically 30 days post-grafting, measurements were conducted weekly. For each combination, three plants were randomly selected to measure the new shoot lengths, shoot thicknesses, internode lengths, and leaf counts, with three repetitions.

### 4.3. Anatomical Observation

To assess the contact surface, the two components of the graft unions from the heterograft and homograft combinations were removed one month post-grafting. The samples were immersed in a 2.5% glutaraldehyde solution within a 0.03 M phosphate buffer for more than 24 h [11], followed by softening in a glycerol–alcohol mixture (1:1) for approximately one week. They were subsequently dehydrated using a sequence of ethanol concentrations (15%, 30%, 50%, 70%, and 95%), with each duration set at 90 min. Following an overnight infiltration in safranin and subsequent decolorization with dimethylbenzene, the specimens were embedded in paraffin. Sections were sliced to a thickness of 10 µm via a sliding microtome. The paraffin sections were ultimately examined using a Zeiss photomicroscope II (Carl Zeiss, Jena, Germany).

### 4.4. Transcriptome Analysis

Total RNA was isolated with the Quick RNA Isolation Kit (Huayueyang, Shanghai, China), following the manufacturer’s guidelines, and subsequently processed with DNase I (TaKaRa, Japan) to eradicate genomic DNA contamination. Subsequently, the mRNA was enriched using oligo (dT) magnetic beads (specifically for eukaryotes). Each library was constructed by combining equal quantities of RNA from three biological replicates at each developmental stage. The techniques, comprising mRNA enrichment, mRNA fragmentation, second-strand cDNA synthesis, size selection, PCR amplification, and subsequent sequencing, were executed at the Beijing Genome Institute (Metware, Wuhan, China). We used FDR < 0.05 and the absolute value of log2Ratio ≥ 1 as the thresholds to judge the significance of the gene expression differences.

Raw reads were filtered to exclude low-quality sequences (including unknown sequences ‘N’), empty tags (sequences comprising solely adaptors), and tags with a single copy number (which may suggest sequencing problems). After the cleaning process, reads were aligned to the reference sequences on 1 March 2024 (http://www.sapindaceae.com/Download.html) utilizing HISAT 2. 0 software (Johns Hopkins University), allowing a maximum of 2 mismatches during the alignment. Tags that corresponded to several gene reference sequences were eliminated, while the remaining tags were considered unequivocal for gene expression analysis. The number of unambiguous tags for each gene was computed and normalized to FPKM (fragments per kilobase of transcript per million mapped reads). Significant differences in genes between groups were identified by the absolute Log_2_FC (fold change).

### 4.5. UPLC–MS/MS-Based Targeted Metabolome Analysis

#### 4.5.1. Chemicals and Reagents

HPLC-grade acetonitrile (ACN) and methanol (MeOH) were acquired from Merck (Darmstadt, Germany). All tests utilized Milli-Q water (Millipore, Bradford, PA, USA). All standards were obtained from Olchemim Ltd. (Olomouc, Czech Republic) and isoReag (Shanghai, China). Acetic acid and formic acid were procured from Sigma-Aldrich (St. Louis, MO, USA). Stock solutions of the standards were formulated at a concentration of 1 mg/mL in methanol. All stock solutions were maintained at −20 °C. Before analysis, the stock solutions were diluted with methanol to produce working solutions.

#### 4.5.2. Sample Preparation and Extraction

A fresh sample was collected, promptly frozen in liquid nitrogen, pulverized (30 Hz, 1 min), and preserved at −80 °C until needed. Fifty milligrams of the plant sample was measured into a 2 mL plastic microtube, cryogenically frozen in liquid nitrogen, and subsequently dissolved in 1 mL of a methanol/water/formic acid mixture (15:4:1, *v*/*v*/*v*). Ten microliters of a mixed solution containing an internal standard at a concentration of 100 ng/mL was included in the extract for measurement purposes. The mixture was vortexed for 10 min and centrifuged for 5 min at 12,000 revolutions per minute, at a temperature of 4 °C. The supernatant was subsequently transferred to sterile plastic microtubes, evaporated to dryness, and reconstituted in 100 μL of 80% methanol (*v*/*v*). Ultimately, it was subjected to filtration using a 0.22 μm membrane filter for further liquid chromatography–mass spectrometry/mass spectrometry analysis [42,43].

#### 4.5.3. UPLC Conditions

The sample extracts were analyzed using a UPLC-ESI-MS/MS system (UPLC, ExionLC™ AD, https://sciex.com.cn/; MS, QTRAP^®^ 6500+, https://sciex.com.cn/) on 1 May 2024. The analytical conditions were as follows: LC: column, Waters ACQUITY UPLC HSS T3 C18 (100 mm × 2.1 mm i.d., 1.8 µm); solvent system, water with 0.04% acetic acid (A), acetonitrile with 0.04% acetic acid (B); gradient program, started at 5% B (0–1 min), increased to 95% B (1–8 min), maintained at 95% B (8–9 min), and finally ramped back to 5% B (9.1–12 min); flow rate, 0.35 mL/min; temperature, 40 °C; injection volume, 2 μL [44,45,46].

#### 4.5.4. ESI-MS/MS Conditions

Linear ion trap (LIT) and triple quadrupole (QQQ) scans were obtained using a triple quadrupole–linear ion trap mass spectrometer (QTRAP), specifically the QTRAP^®^ 6500+ LC-MS/MS System, which features an ESI Turbo Ion-Spray interface. The instrument operated in both positive and negative ion modes and was managed using Analyst 1.6.3 software (Sciex). The parameters for the ESI source operation were as follows: ion source: ESI+/−; source temperature: 550 °C; ion spray voltage (IS): 5500 V (positive), −4500 V (negative); curtain gas (CUR) set at 35 psi. Phytohormones were examined utilizing planned multiple reaction monitoring (MRM). Data acquisitions were performed using Analyst 1.6.3 software (Sciex). Multiquant 3.0.3 software (Sciex) was employed to quantify all metabolites. The parameters of the mass spectrometer, including the declustering potentials (DPs) and collision energies (CEs) for certain MRM transitions, were optimized further for the DP and CE. Each period involved the monitoring of a distinct set of MRM transitions based on the metabolites eluted during that interval [47,48,49].

Phytohormone levels were quantified by MetWare (http://www.metware.cn/) on 1 April 2024 utilizing AB Sciex QTRAP 6500 LC-MS/MS equipment. The method for calculating the solid sample content was as follows: We determined the concentration (ng/mL) by substituting the peak area ratio of the sample into the standard calibration curve. Then, we calculated the hormone content (ng/g) using the following formula: hormone content (ng/g) = (c × V)/(1000 × m) (c: concentration obtained from the standard curve (ng/mL), V: reconstitution volume (μL), m: sample mass (g)). Unit conversions were already incorporated into the formula; we simply input the numerical values for calculation. The results of the hierarchical cluster analysis (HCA) for samples and metabolites were displayed as heatmaps accompanied by dendrograms. A hierarchical cluster analysis (HCA) was conducted using the R package pheatmap. For the HCA, the normalized signal intensities of metabolites (unit variance scaling) are represented as a color spectrum. The regulated metabolites between groups were identified using the absolute Log2FC (fold change). The identified metabolites were annotated utilizing the KEGG compound database (http://www.kegg.jp/kegg/compound/) in 1 July 2024, and the annotated metabolites were subsequently mapped to the KEGG pathway database (http://www.kegg.jp/kegg/pathway.html) in 1 July 2024. Pathways with highly regulated metabolites were subsequently analyzed using MSEA (metabolite sets enrichment analysis), with their significance assessed by *p*-values derived from the hypergeometric test.

### 4.6. Quantitative Real-Time PCR Analysis

Total RNA was extracted as previously outlined and reverse transcribed utilizing oligo (dT) primers and M-MLV reverse transcriptase (Invitrogen, Waltham, MA, USA), following the manufacturer’s instructions. The transcript levels were evaluated using quantitative RT-PCR with the DyNAmo Flash SYBR Green qPCR kit (Thermo, Waltham, MA, USA) and the CFX96 qPCR System (Bio-Rad, Hercules, CA, USA), following the manufacturer’s instructions. Each response was conducted in triplicate using three biological replicates. The Ct values for the LcActin gene (HQ615689) served to normalize the data. The 2^ΔΔCt^ [50] method was employed to assess the unigene expression levels. The primer list is provided in Appendix A.

### 4.7. Statistical Analysis

The SPSS program (version 22.0, IBM Corp., Armonk, NY, USA) was employed to analyze the least significant difference (LSD) at the 5% significance threshold. Graphing was conducted using OriginLab (version 2019b, OriginLab Inc., Northampton, MA, USA).

## 5. Conclusions

This research emphasizes the essential function of auxin transport, secondary metabolite pathways, and hormone signaling in the compatibility of litchi grafts. The effective grafting of ‘FZX’ scions onto ‘ZNX’ rootstocks was significantly linked to improved auxin transport, the heightened expression of essential auxin-related genes, and the upregulation of secondary metabolite biosynthesis. The results indicate that auxin and salicylic acid are crucial during the initial stage of grafting, whereas additional phytohormones are essential for subsequent stages to guarantee proper graft development. The transcriptome analysis yielded significant insights into the genetic foundations of grafting success, identifying prospective molecular targets for improving graft compatibility. Future research should concentrate on clarifying the molecular mechanisms of auxin transport and secondary metabolism in litchi grafting to enhance grafting techniques for increased horticultural yield.

## Figures and Tables

**Figure 1 ijms-26-04231-f001:**
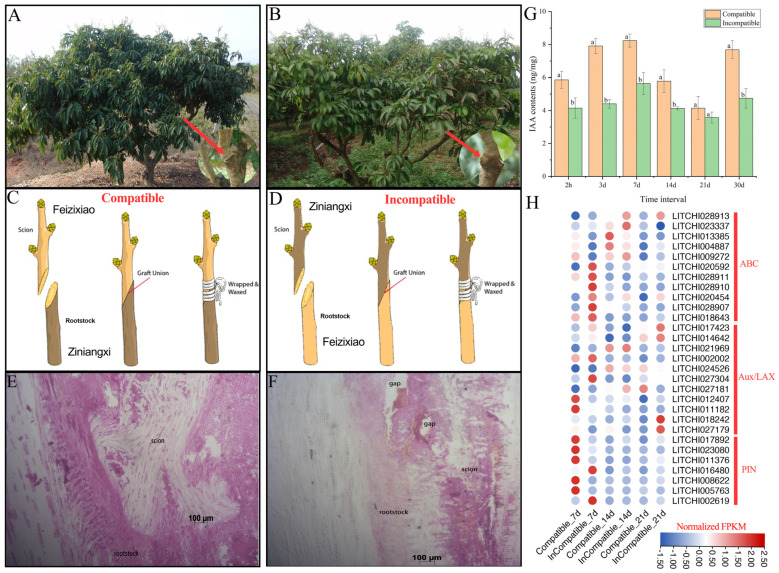
Performance in the field, an anatomical analysis, and a gene expression analysis related to compatible and incompatible litchi grafting. (**A**,**B**) The performance of grafted litchi trees in the field. The pictures were taken 30 days post-grafting. (**C**,**D**) An illustration of compatible and incompatible litchi grafting. For compatible litchi grafting, cultivar FZX was used as the ‘scion’ and ZNX as the ‘rootstock’ with the opposite for incompatible grafting. Red arrow indicating the graft union. (**E**) An anatomical analysis of the compatible and (**F**) incompatible litchi grafts. The bar size indicates 100 μm. The anatomical analysis was performed 30 days post-grafting. (**G**) Total endogenous indole acetic acid (IAA) contents from compatible and incompatible litchi grafts. Error bars indicate the standard error of the mean (SEM) of three biological replicates. Distinct lowercase letters signify a statistically significant difference at the *p*-value 0.05 level. The analysis was performed 2 h and 3, 7, 14, 21, and 30 days post-grafting. Lowercase letters denote statistically significant differences between three biological replicates. (**H**) A heatmap highlighting the expression analysis of the ABC transporter, Aux/LAX, and PIN genes in the compatible and incompatible litchi grafts. The expression was observed at 7, 14, and 21 days post-grafting in compatible and incompatible litchi grafts.

**Figure 2 ijms-26-04231-f002:**
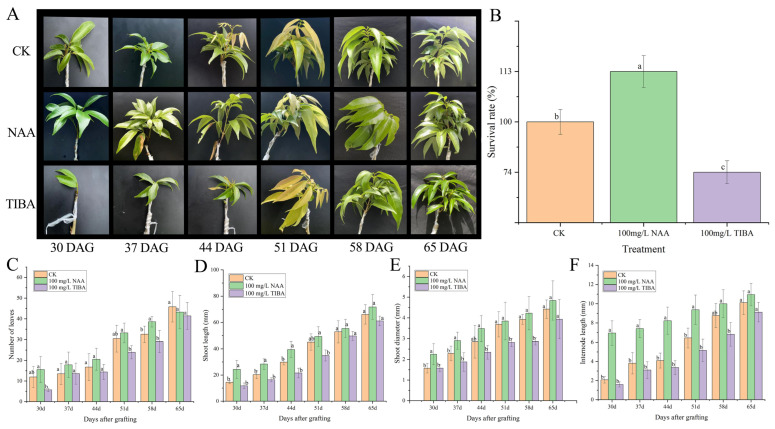
Evaluation of the performance and physiological characteristics of litchi grafts in the field following the application of 100 mg/L NAA and 100 mg/L TIBA. (**A**) The field performance of compatible litchi grafts at 30, 37, 44, 51, 58, and 65 days post-grafting (DAG). (**B**) The survival rates (%) of litchi grafts in the control, 100 mg/L NAA, and 100 mg/L TIBA treatments. (**C**) The number of leaves, (**D**) shoot length (mm), (**E**) shoot diameter (mm), and (**F**) internode length (mm) in the control, 100 mg/L NAA-treated, and 100 mg/L TIBA-treated grafts. The analysis was conducted 30, 37, 44, 51, 58, and 65 days after germination (DAG). Distinct lowercase letters signify a statistically significant difference at the *p*-value 0.05 level. The data are presented as the standard error of the mean from three to five biological replicates.

**Figure 3 ijms-26-04231-f003:**
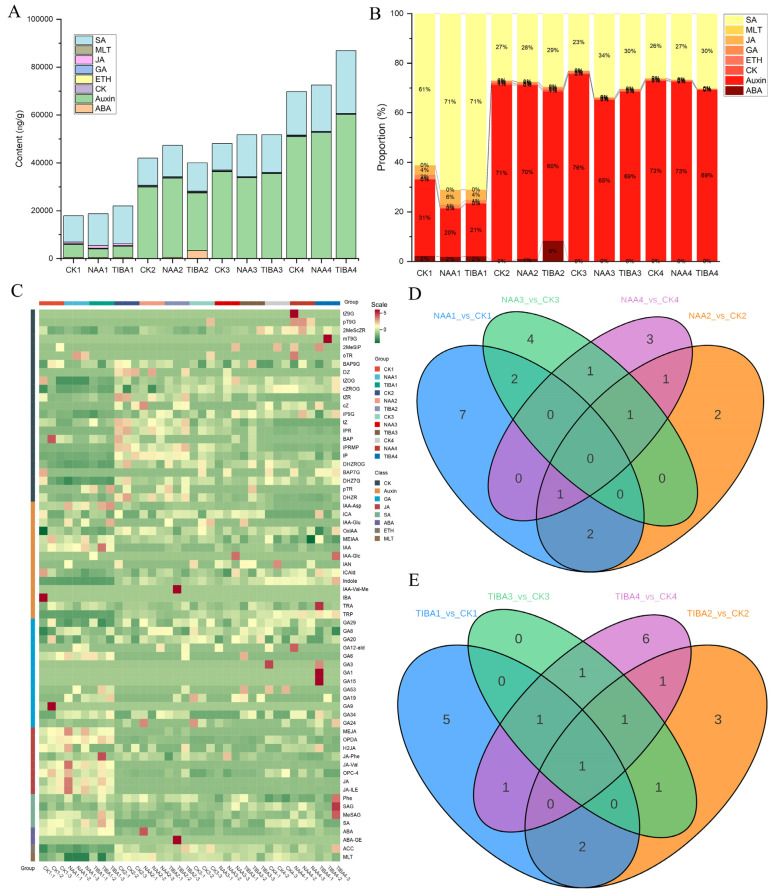
A hormonal metabolomic investigation of litchi grafts under control conditions following 100 mg/L NAA and 100 mg/L TIBA application. (**A**) The total contents and (**B**) percentages of multiple hormones in the litchi grafts following treatment with 100 mg/L NAA, 100 mg/L TIBA, and the control. (**C**) A heatmap depicting the expression patterns of hormone metabolites. (**D**) A Venn diagram illustrating the differentially expressed hormones among CK1 vs. NAA1, CK2 vs. NAA2, CK3 vs. NAA3, and CK4 vs. NAA4. (**E**) A Venn diagram illustrating the variably expressed hormones in the litchi grafts, comparing CK1 with TIBA1, CK2 with TIBA2, CK3 with TIBA3, and CK4 with TIBA4. CK1, CK2, CK3, and CK4 represent control sample collections at 2 hours, 3 days, 7 days, and 14 days post-grafting, respectively; analogous naming conventions apply for NAA and TIBA. SA: salicylic acid, MLT: melatonin, JA: jasmonic acid, GA: gibberellic acid, ETH: ethylene, CK: cytokinin, ABA: abscisic acid, Auxin: auxin.

**Figure 4 ijms-26-04231-f004:**
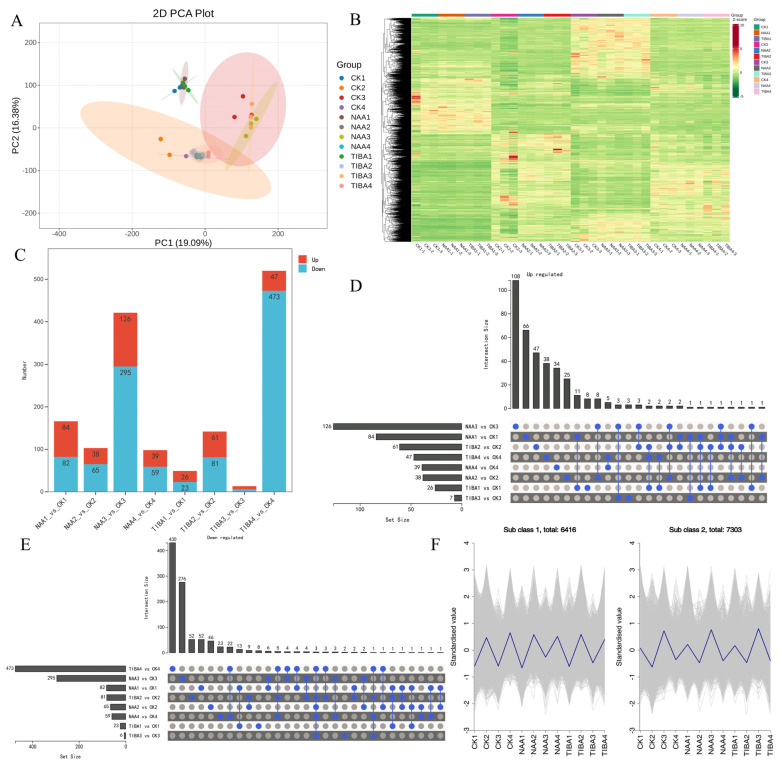
Assessments of RNA sequencing and DEGs. (**A**) PCA score plot illustrating 36 transcriptome samples. (**B**) Heat map depicting relative expression of all genes using Log_2_FPKM. (**C**) Abundance of DEGs at all phases. (**D**) UpSet R figure illustrating the numbers of upregulated (**E**) and downregulated DEGs identified throughout all stages. (**F**) K-means clustering investigation of DEGs across all phases.

**Figure 5 ijms-26-04231-f005:**
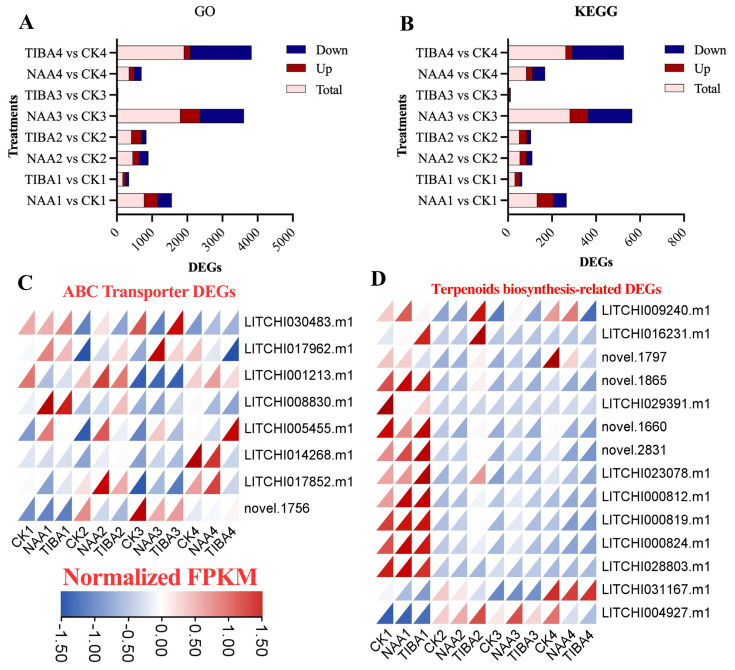
A visualization of the DEGs detected in the GO and KEGG databases, along with the DEGs associated with secondary metabolites. (**A**) DEGs identified in the gene ontology and (**B**) KEGG pathways. (**C**,**D**) Heatmaps illustrating the expression levels of ABC transporter- and terpenoid biosynthesis-associated genes discovered among the DEGs. The color scheme employed in this study signifies varying levels of gene expression. Blue signifies low expression, whereas red denotes high expression.

**Figure 6 ijms-26-04231-f006:**
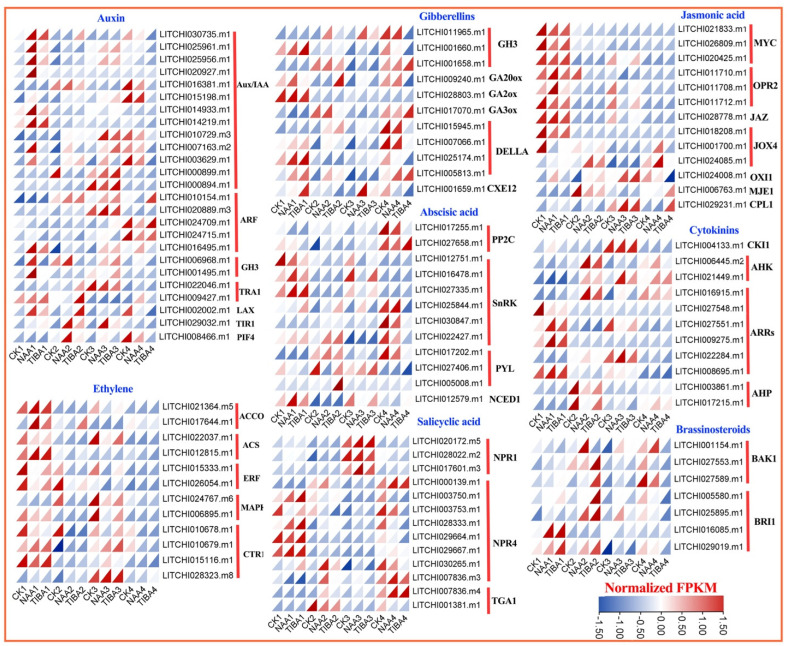
Expression profiles of the differentially expressed genes associated with plant hormone biosynthesis and signal transduction. The color scheme employed in this study signifies varying levels of gene expression, with blue signifying low expression and red denoting high expression.

**Figure 7 ijms-26-04231-f007:**
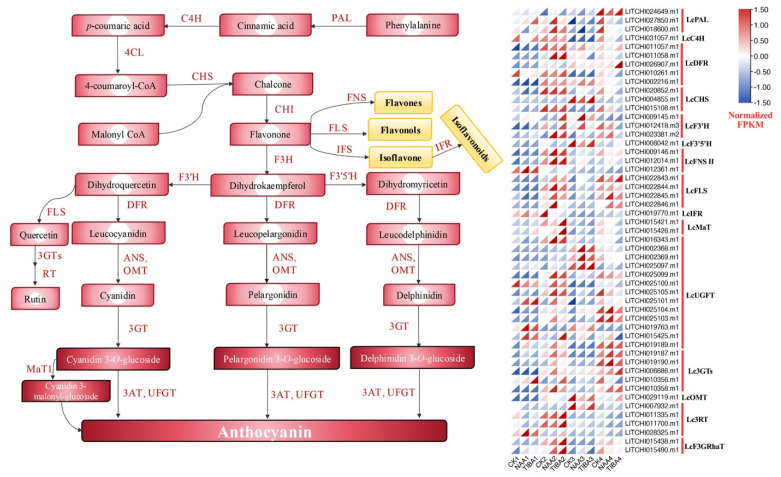
Expression patterns of DEGs associated with the anthocyanin and phenylpropanoid biosynthesis pathways. The expression profiles of the hormonal DEGs are represented in a heatmap, with red marking the highest expression and blue indicating the lowest.

**Figure 8 ijms-26-04231-f008:**
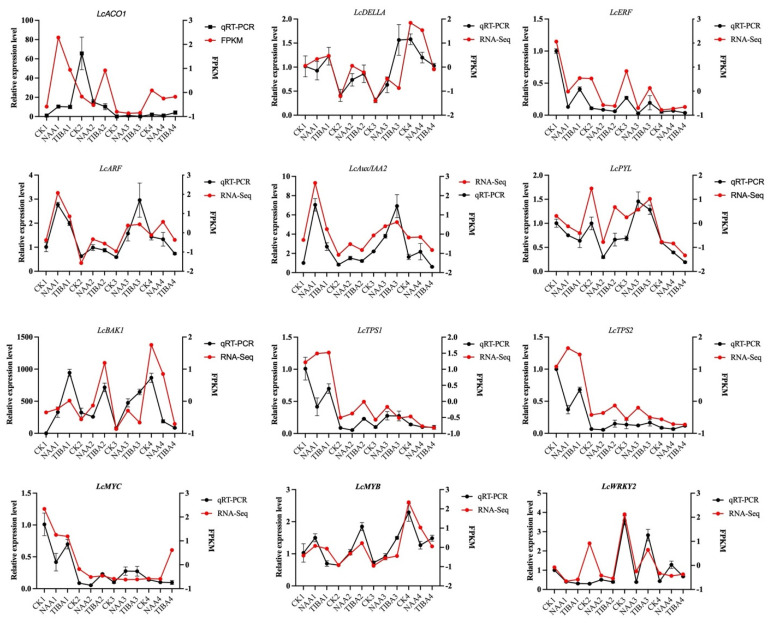
Quantitative RT-PCR validation of 12 genes, together with their FPKM values. Actin served as an internal control. The error bars denote the mean ± SEM (n = 3).

**Figure 9 ijms-26-04231-f009:**
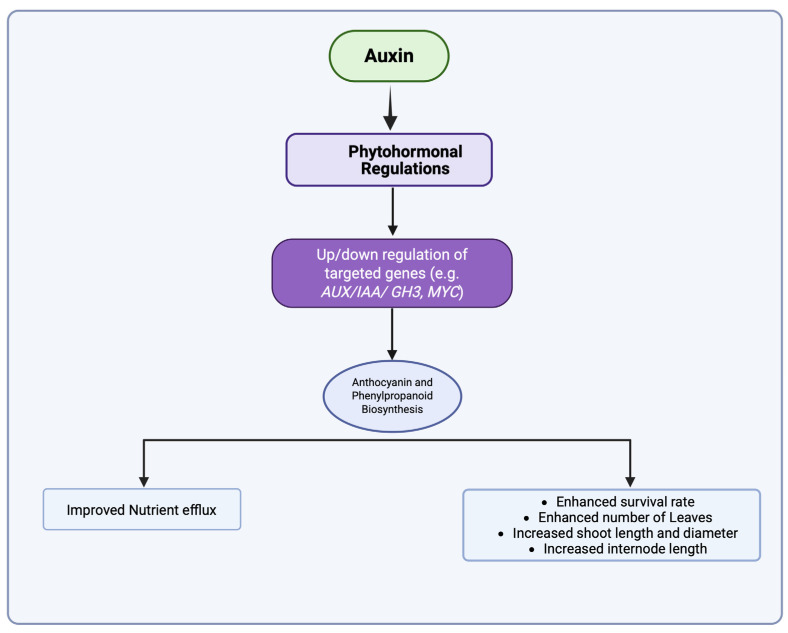
A schematic representation of auxin’s effect on litchi grafting.

## Data Availability

The transcriptome raw data have been submitted to the SRA database of the NCBI (BioProject ID: PRJNA1231586).

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
