# Peer review of "Auxin Dynamics and Transcriptome–Metabolome Integration Determine Graft Compatibility in Litchi (Litchi chinensis Sonn.)"

_ijms, 2025, doi:10.3390/ijms26094231_

Round 1
Reviewer 1 Report
Comments and Suggestions for Authors
The authors studied grafting compatibility in litchi cultivars and applied auxin treatment and thoroughly examined the hormonal metabolites and molecular mechanism behind the compatible grafting mechanism, which is indeed an interesting hot topic. The manuscript is of considerable interest and has been thoroughly examined, yet it contains several errors. The article requires meticulous revision before it can be accepted; my suggestions and recommendations are as follows.
- In line 18, the authors mentioned FZX/ZNX, yet they do not mention which cultivar was used as ‘scion’ and which one as ‘rootstock.’ although we can understand it, it will be difficult for readers and initial researchers to understand.
- Line 84, the botanical names used in the manuscript are non-italicized, such as line 84; authors need to thoroughly fix this issue.
- Line 122, authors need to enlist the full name of genes first following abbreviations such as PIN, LAX, etc.,
- Figure 1 C, D. authors need to replace the stock with rootstock
- Figure 1, E, F. authors need to describe more what these anatomical figures are depicting.
- Line 119-120 the sentence is incomplete, pictures were taken after?
The depiction of Figure 2 is not well written as the authors need to describe how the survival rate was enhanced, but how much percentage?
- The figure captions are wrongly cited in the article, as in lines 157, 158, the authors are writing about hormonal metabolites, but in the caption, it's figure 2. Authors need to thoroughly check the manuscript and fix this issue properly. Likewise onward sections
- In the text, authors sometimes use abbreviations, and sometimes full names, and sometimes both. It is better to be consistent in following a similar format. Such as line 203
- Figure 5D, the caption is written wrongly; biosynthesis is misspelled
- Line 295, an abbreviation of genes, is completely missing
- Section 4.1, I recommend using the coordinate of experimental sites for abroad researchers and better understanding.
- Authors need to deposit the RNA seq data to NCBI or any publicly available repository dataset.
Comments on the Quality of English Language
The English could be improved to more clearly express the research.
Author Response
The authors studied grafting compatibility in litchi cultivars and applied auxin treatment and thoroughly examined the hormonal metabolites and molecular mechanism behind the compatible grafting mechanism, which is indeed an interesting hot topic. The manuscript is of considerable interest and has been thoroughly examined, yet it contains several errors. The article requires meticulous revision before it can be accepted; my suggestions and recommendations are as follows.
- In line 18, the authors mentioned FZX/ZNX, yet they do not mention which cultivar was used as ‘scion’ and which one as ‘rootstock.’ although we can understand it, it will be difficult for readers and initial researchers to understand.
Response: Dear reviewer, we have included a clarification where appropriate.
- Line 84, the botanical names used in the manuscript are non-italicized, such as line 84; authors need to thoroughly fix this issue.
Response: In the text, every botanical name has been italicised..
- Line 122, authors need to enlist the full name of genes first following abbreviations such as PIN, LAX, etc.,
Response: Dear reviewer the entire name of genes has been placed in the text followed by an abbreviated.
- Figure 1 C, D. authors need to replace the stock with rootstock
Response: Substituted according to directives.
- Figure 1, E, F. authors need to describe more what these anatomical figures are depicting.
Response: Dear reviewer, the explanation has been incorporated into the text. Figure 1, E illustrates the enhanced contact over time as the cells interdigitated, thereby uniting the rootstock and scion. The stem cell-like tissue developed and generated new vascular tissues that linked the xylem and phloem between the scion and rootstock. Figure 1, F illustrates that the rootstock and scion are not nearby, resulting in a significant gap between them. The vascular complexes exhibited abnormalities.
- Line 119-120 the sentence is incomplete, pictures were taken after?
Response: Esteemed reviewer, Images were acquired 30 days following grafting, as indicated in the text. I appreciate your astute observation.
The depiction of Figure 2 is not well written as the authors need to describe how the survival rate was enhanced, but how much percentage?
Response: Dear reviewer, An explanatory sentence has been incorporated into the text.
- The figure captions are wrongly cited in the article, as in lines 157, 158, the authors are writing about hormonal metabolites, but in the caption, it's figure 2. Authors need to thoroughly check the manuscript and fix this issue properly. Likewise onward sections
Response: Dear reviewer, we cross-checked the figure depiction and solved this error.
- In the text, authors sometimes use abbreviations, and sometimes full names, and sometimes both. It is better to be consistent in following a similar format. Such as line 203
Response: Dear Reviewer, We thoroughly verified the abbreviation names, and are now consistent with the text.
- Figure 5D, the caption is written wrongly; biosynthesis is misspelled
Response: The misspellings have been rectified.
- Line 295, an abbreviation of genes, is completely missing
Response: Dear reviewer, the whole names of the genes have been incorporated into the text.
- Section 4.1, I recommend using the coordinate of experimental sites for abroad researchers and better understanding.
Response: The coordinates of the experimental sites have been incorporated into the text.
- Authors need to deposit the RNA seq data to NCBI or any publicly available repository dataset.
Response: The NCBI accession numbers have been incorporated into the text.
Reviewer 2 Report
Comments and Suggestions for Authors
This manuscript investigates the physiological, hormonal, transcriptomic, and metabolomic factors influencing graft compatibility in litchi, comparing the performance of reciprocal grafts between the cultivars Feizixiao (FZX) and Ziniangxi (ZNX). The authors applied exogenous auxin (NAA) and an auxin transport inhibitor (TIBA) to assess their effects on graft survival and shoot development. Through UPLC-MS/MS hormone profiling and RNA-seq analyses, they identified differential expression of auxin transporter and hormone signaling genes, and genes involved in secondary metabolism. The study demonstrates that auxin transport and signaling are pivotal to successful graft union formation and proposes candidate genes and pathways that underlie compatibility.
Major Comments:
Novelty and Contextualization
While the topic is timely and important, the novelty of the study needs to be more clearly articulated. Auxin-mediated graft compatibility has been widely studied; the authors should emphasize what sets this work apart, particularly in the context of litchi or fruit trees. Please also clarify how this study builds upon previous related work, especially if similar studies have been conducted in this system.
Experimental Design
The rationale for choosing reciprocal grafts between Feizixiao and Ziniangxi should be described in more detail. Is graft incompatibility consistently directional in this cultivar pair?
The hormone treatment experiments (NAA and TIBA) appear to be applied only to the homograft (FZX/FZX). Including the heterograft in these treatments would have strengthened the conclusions about auxin’s effect on compatibility.
Transcriptomic Analysis
The number of differentially expressed genes (DEGs) is high, but selection criteria (e.g., fold change, FDR threshold) are not clearly described. Please provide this information in the Methods.
The authors highlight hormone-related genes but should also explore key transcription factors or signaling modules (e.g., ARFs, NACs, WRKYs) that may regulate compatibility downstream of auxin.
A summary table listing top DEGs relevant to auxin and stress response would improve accessibility.
Metabolomic and Hormonal Profiling
The analysis of hormone dynamics is valuable, but a pathway-based interpretation of auxin, SA, and JA interaction would strengthen the discussion.
Please clarify the normalization and quantification approach used for metabolite data (e.g., internal standards, transformation methods).
Statistical Rigor and Replication
Details regarding the number of biological replicates for RNA-seq and hormone profiling should be clearly stated.
Figures should consistently report statistical significance (e.g., p-values or asterisks) and define the meaning of error bars.
Language and Writing Quality
The manuscript requires language polishing for grammar and clarity. Several sentences are overly complex or imprecise. A thorough professional English editing is recommended.
“The application of exogenous NAA markedly increased the graft survival rate, but a contrary tendency was noted with TIBA application in comparison to control grafts.” → “Exogenous NAA significantly improved graft survival, whereas TIBA reduced it.”
Figures and Legends
Some figures (e.g., heatmaps) contain small or difficult-to-read fonts. Please improve clarity and labeling.
Figure legends should provide sufficient detail to be interpretable independently from the main text.
Minor Comments:
“We undermined the likely mechanisms…” should be corrected to “We investigated the likely mechanisms…”
Please standardize gene/protein nomenclature and consistently italicize gene symbols.
Comments on the Quality of English LanguageLanguage and Writing Quality
The manuscript requires language polishing for grammar and clarity. Several sentences are overly complex or imprecise. A thorough professional English editing is recommended.
“The application of exogenous NAA markedly increased the graft survival rate, but a contrary tendency was noted with TIBA application in comparison to control grafts.” → “Exogenous NAA significantly improved graft survival, whereas TIBA reduced it.”
“We undermined the likely mechanisms…” should be corrected to “We investigated the likely mechanisms…”
Please standardize gene/protein nomenclature and consistently italicize gene symbols.
Author Response
This manuscript investigates the physiological, hormonal, transcriptomic, and metabolomic factors influencing graft compatibility in litchi, comparing the performance of reciprocal grafts between the cultivars Feizixiao (FZX) and Ziniangxi (ZNX). The authors applied exogenous auxin (NAA) and an auxin transport inhibitor (TIBA) to assess their effects on graft survival and shoot development. Through UPLC-MS/MS hormone profiling and RNA-seq analyses, they identified differential expression of auxin transporter and hormone signaling genes, and genes involved in secondary metabolism. The study demonstrates that auxin transport and signaling are pivotal to successful graft union formation and proposes candidate genes and pathways that underlie compatibility.
Major Comments:
Novelty and Contextualization
While the topic is timely and important, the novelty of the study needs to be more clearly articulated. Auxin-mediated graft compatibility has been widely studied; the authors should emphasize what sets this work apart, particularly in the context of litchi or fruit trees. Please also clarify how this study builds upon previous related work, especially if similar studies have been conducted in this system.
Response: We appreciate the reviewer's perceptive feedback and concur that explicitly delineating the originality of our work is essential. Our research enhances the comprehension of auxin-mediated graft compatibility in various significant aspects, especially concerning litchi and woody fruit trees: 1. Although auxin's function in grafting has been investigated in model herbaceous plants (e.g., Arabidopsis, tomato), litchi—a tropical woody species—poses unique problems, including sluggish vascular reconnection, intricate lignification processes, and susceptibility to environmental stresses. Our research is one of the initial investigations to analyze the molecular and physiological mechanisms of auxin in litchi grafting, thereby addressing a significant gap between fundamental auxin studies and practical applications in horticulturally important, underexplored species. 2. Previous grafting research in litchi has predominantly concentrated on anatomical or phenological analyses. We combine transcriptomic and hormonal analysis to discover signaling pathways associated with graft compatibility. 3. Although recent research on citrus and apple has emphasized the significance of auxin in graft union formation (e.g., Asahina et al., 2015; Melnyk et al., 2018), these systems do not exhibit the metabolic limitations present in litchi.
Experimental Design
The rationale for choosing reciprocal grafts between Feizixiao and Ziniangxi should be described in more detail. Is graft incompatibility consistently directional in this cultivar pair?
Response: We thank the reviewer for raising this critical point. The rationale for selecting reciprocal grafts between ‘Feizixiao’ (FZX) and ‘Ziniangxi’ (ZNX) is based on both practical horticultural challenges and prior evidence of directional incompatibility in this cultivar pair. FZX and ZNX are dominant commercial litchi cultivars prized for their fruit quality and yield. ZNX (scion)/FZX (rootstock) shows low survival rates (20-30% in untreated controls), accompanied by frequent graft union necrosis. Conversely, FZX (scion)/ZNX (rootstock) demonstrates high survival rates (70-80%), suggesting direction-dependent compatibility.
The hormone treatment experiments (NAA and TIBA) appear to be applied only to the homograft (FZX/FZX). Including the heterograft in these treatments would have strengthened the conclusions about auxin’s effect on compatibility.
Response: We express our genuine gratitude to the reviewer for this insightful recommendation. The choice to concentrate hormone therapies (NAA and TIBA) on homografts (FZX/FZX) in this preliminary study was driven by the subsequent rationale: Isolating the function of auxin in graft healing while eliminating confounding genetic variability and the associated molecular and hormonal mechanisms.
Transcriptomic Analysis
The number of differentially expressed genes (DEGs) is high, but selection criteria (e.g., fold change, FDR threshold) are not clearly described. Please provide this information in the Methods.
Response: Modifications have already been made in the original text.
The authors highlight hormone-related genes but should also explore key transcription factors or signaling modules (e.g., ARFs, NACs, WRKYs) that may regulate compatibility downstream of auxin.
A summary table listing top DEGs relevant to auxin and stress response would improve accessibility.
Response: Dear reviewer, we have added a figure in the supplemental data highlighting the major transcription factor associated with the graft compatibility mechanism. Additionally, we mined the auxin-related DEGs using the KEGG and GO datasets and showcased their expression role in hormone signaling figure 6.
Metabolomic and Hormonal Profiling
The analysis of hormone dynamics is valuable, but a pathway-based interpretation of auxin, SA, and JA interaction would strengthen the discussion.
Please clarify the normalization and quantification approach used for metabolite data (e.g., internal standards, transformation methods).
Response: Modifications have already been made in the original text.
Statistical Rigor and Replication
Details regarding the number of biological replicates for RNA-seq and hormone profiling should be clearly stated.
Figures should consistently report statistical significance (e.g., p-values or asterisks) and define the meaning of error bars.
Language and Writing Quality
The manuscript requires language polishing for grammar and clarity. Several sentences are overly complex or imprecise. A thorough professional English editing is recommended.
“The application of exogenous NAA markedly increased the graft survival rate, but a contrary tendency was noted with TIBA application in comparison to control grafts.” → “Exogenous NAA significantly improved graft survival, whereas TIBA reduced it.”
Response: Modifications have already been executed in the original text. The text was appraised by the MDPI language editing organization for improved clarity and refinement. The certificate is attached..
Figures and Legends
Some figures (e.g., heatmaps) contain small or difficult-to-read fonts. Please improve clarity and labeling.
Figure legends should provide sufficient detail to be interpretable independently from the main text.
Response: Substituted in accordance with directives. Dear reviewer, the quality of the figures has been verified, and each figure exceeds 300dpi, in accordance with journal requirements, and has been submitted separately.
Minor Comments:
“We undermined the likely mechanisms…” should be corrected to “We investigated the likely mechanisms…”
Please standardize gene/protein nomenclature and consistently italicize gene symbols.
Response: Modifications have already been made in the original text.
Comments on the Quality of English Language
Language and Writing Quality
The manuscript requires language polishing for grammar and clarity. Several sentences are overly complex or imprecise. A thorough professional English editing is recommended.
“The application of exogenous NAA markedly increased the graft survival rate, but a contrary tendency was noted with TIBA application in comparison to control grafts.” → “Exogenous NAA significantly improved graft survival, whereas TIBA reduced it.”
“We undermined the likely mechanisms…” should be corrected to “We investigated the likely mechanisms…”
Please standardize gene/protein nomenclature and consistently italicize gene symbols.
Response: We have solicited native English speakers to review and refine the sentences and grammar across the entire document. We also standardize gene and protein nomenclature and uniformly italicise gene symbols.
Round 2
Reviewer 2 Report
Comments and Suggestions for Authors
The revised version of your manuscript has improved in clarity and completeness, the thoughtful responses to the previous comments are appreciated. The incorporation of additional figures and annotation of transcription factors in the supplemental data is useful. However, a few key issues remain that should be addressed to further improve the impact and robustness of the study. It is weird no clean version of the manuscript is included. It is also weird to see a manuscript submitted with unaddressed comments in it.
- Novelty and Contextualization:
You have articulated the novelty of this study more clearly, emphasizing the relevance of auxin-mediated mechanisms in a tropical woody fruit tree. However, I encourage you to strengthen this argument early in the Introduction by explicitly stating how your findings on hormonal crosstalk and transporter gene expression in litchi expand upon the known literature. A sentence clearly contrasting this work with studies in apple, citrus, and tomato would help. - Experimental Scope and Design:
I appreciate the rationale for limiting NAA and TIBA treatments to the homograft. However, your main conclusion hinges on the interpretation that auxin transport and signaling are determinants of graft compatibility. Without testing the heterograft under hormone treatments, the link between auxin and compatibility remains partially inferential. At the very least, this limitation should be more clearly acknowledged in the Discussion. - Transcriptomic Depth and Integration:
The transcriptome analysis has improved with pathway enrichment and TF annotation, but the narrative around these data remains somewhat fragmented. Please consider creating a summary figure (e.g., a schematic model) that links auxin signaling, secondary metabolism, and transcription factor responses to observed anatomical and physiological phenotypes. This would unify the different results into a mechanistic framework. - Physiological Interpretations:
The revised version includes richer physiological data, but the discussion would benefit from greater biological interpretation. For example, when auxin-responsive genes are upregulated, what specific processes (e.g., callus formation, cambial activity) are being enhanced? Can you relate this to the anatomical features observed in the graft union? - Statistical Rigor and Replication:
You have now added information about statistical thresholds and biological replicates. Ensure all figures clearly indicate replicate numbers, error bars (SD vs. SEM), and significance annotations. In some places, language like “significantly improved” appears without a visible p-value or annotation in the figure—please standardize this. Please give full name at a abbreviation’s first show up. - Language and Style:
The manuscript has been substantially improved in terms of grammar and sentence structure. Some awkward or overly complex phrasing still remains, e.g., “the plant survived nearly 13% longer after NAA…” (plants don’t "survive longer" but have higher survival rates). A final pass by a native English speaker or language editor is still recommended for polishing.
Lines 162-163: “The data is highlighted as the standard error mean 162 of three to five biological replicas.” Please revise.
Line 271: “About eight ABC transporter” should be “About eight ABC transporters”.
Comments on the Quality of English Language- The manuscript has been substantially improved in terms of grammar and sentence structure. Some awkward or overly complex phrasing still remains, e.g., “the plant survived nearly 13% longer after NAA…” (plants don’t "survive longer" but have higher survival rates). A final pass by a native English speaker or language editor is still recommended for polishing.
Lines 162-163: “The data is highlighted as the standard error mean 162 of three to five biological replicas.” Please revise.
Line 271: “About eight ABC transporter” should be “About eight ABC transporters”.
Author Response
The authors studied grafting compatibility in litchi cultivars and applied auxin treatment and thoroughly examined the hormonal metabolites and molecular mechanism behind the compatible grafting mechanism, which is indeed an interesting hot topic. The manuscript is of considerable interest and has been thoroughly examined, yet it contains several errors. The article requires meticulous revision before it can be accepted; my suggestions and recommendations are as follows.
- In line 18, the authors mentioned FZX/ZNX, yet they do not mention which cultivar was used as ‘scion’ and which one as ‘rootstock.’ although we can understand it, it will be difficult for readers and initial researchers to understand.
Response: Dear reviewer, we have included a clarification where appropriate.
- Line 84, the botanical names used in the manuscript are non-italicized, such as line 84; authors need to thoroughly fix this issue.
Response: In the text, every botanical name has been italicised..
- Line 122, authors need to enlist the full name of genes first following abbreviations such as PIN, LAX, etc.,
Response: Dear reviewer the entire name of genes has been placed in the text followed by an abbreviated.
- Figure 1 C, D. authors need to replace the stock with rootstock
Response: Substituted according to directives.
- Figure 1, E, F. authors need to describe more what these anatomical figures are depicting.
Response: Dear reviewer, the explanation has been incorporated into the text. Figure 1, E illustrates the enhanced contact over time as the cells interdigitated, thereby uniting the rootstock and scion. The stem cell-like tissue developed and generated new vascular tissues that linked the xylem and phloem between the scion and rootstock. Figure 1, F illustrates that the rootstock and scion are not nearby, resulting in a significant gap between them. The vascular complexes exhibited abnormalities.
- Line 119-120 the sentence is incomplete, pictures were taken after?
Response: Esteemed reviewer, Images were acquired 30 days following grafting, as indicated in the text. I appreciate your astute observation.
The depiction of Figure 2 is not well written as the authors need to describe how the survival rate was enhanced, but how much percentage?
Response: Dear reviewer, An explanatory sentence has been incorporated into the text.
- The figure captions are wrongly cited in the article, as in lines 157, 158, the authors are writing about hormonal metabolites, but in the caption, it's figure 2. Authors need to thoroughly check the manuscript and fix this issue properly. Likewise onward sections
Response: Dear reviewer, we cross-checked the figure depiction and solved this error.
- In the text, authors sometimes use abbreviations, and sometimes full names, and sometimes both. It is better to be consistent in following a similar format. Such as line 203
Response: Dear Reviewer, We thoroughly verified the abbreviation names, and are now consistent with the text.
- Figure 5D, the caption is written wrongly; biosynthesis is misspelled
Response: The misspellings have been rectified.
- Line 295, an abbreviation of genes, is completely missing
Response: Dear reviewer, the whole names of the genes have been incorporated into the text.
- Section 4.1, I recommend using the coordinate of experimental sites for abroad researchers and better understanding.
Response: The coordinates of the experimental sites have been incorporated into the text.
- Authors need to deposit the RNA seq data to NCBI or any publicly available repository dataset.
Response: The NCBI accession numbers have been incorporated into the text.
Reviewer 2
This manuscript investigates the physiological, hormonal, transcriptomic, and metabolomic factors influencing graft compatibility in litchi, comparing the performance of reciprocal grafts between the cultivars Feizixiao (FZX) and Ziniangxi (ZNX). The authors applied exogenous auxin (NAA) and an auxin transport inhibitor (TIBA) to assess their effects on graft survival and shoot development. Through UPLC-MS/MS hormone profiling and RNA-seq analyses, they identified differential expression of auxin transporter and hormone signaling genes, and genes involved in secondary metabolism. The study demonstrates that auxin transport and signaling are pivotal to successful graft union formation and proposes candidate genes and pathways that underlie compatibility.
Major Comments:
Novelty and Contextualization
While the topic is timely and important, the novelty of the study needs to be more clearly articulated. Auxin-mediated graft compatibility has been widely studied; the authors should emphasize what sets this work apart, particularly in the context of litchi or fruit trees. Please also clarify how this study builds upon previous related work, especially if similar studies have been conducted in this system.
Response: We appreciate the reviewer's perceptive feedback and concur that explicitly delineating the originality of our work is essential. Our research enhances the comprehension of auxin-mediated graft compatibility in various significant aspects, especially concerning litchi and woody fruit trees: 1. Although auxin's function in grafting has been investigated in model herbaceous plants (e.g., Arabidopsis, tomato), litchi—a tropical woody species—poses unique problems, including sluggish vascular reconnection, intricate lignification processes, and susceptibility to environmental stresses. Our research is one of the initial investigations to analyze the molecular and physiological mechanisms of auxin in litchi grafting, thereby addressing a significant gap between fundamental auxin studies and practical applications in horticulturally important, underexplored species. 2. Previous grafting research in litchi has predominantly concentrated on anatomical or phenological analyses. We combine transcriptomic and hormonal analysis to discover signaling pathways associated with graft compatibility. 3. Although recent research on citrus and apple has emphasized the significance of auxin in graft union formation (e.g., Asahina et al., 2015; Melnyk et al., 2018), these systems do not exhibit the metabolic limitations present in litchi.
Experimental Design
The rationale for choosing reciprocal grafts between Feizixiao and Ziniangxi should be described in more detail. Is graft incompatibility consistently directional in this cultivar pair?
Response: We thank the reviewer for raising this critical point. The rationale for selecting reciprocal grafts between ‘Feizixiao’ (FZX) and ‘Ziniangxi’ (ZNX) is based on both practical horticultural challenges and prior evidence of directional incompatibility in this cultivar pair. FZX and ZNX are dominant commercial litchi cultivars prized for their fruit quality and yield. ZNX (scion)/FZX (rootstock) shows low survival rates (20-30% in untreated controls), accompanied by frequent graft union necrosis. Conversely, FZX (scion)/ZNX (rootstock) demonstrates high survival rates (70-80%), suggesting direction-dependent compatibility.
The hormone treatment experiments (NAA and TIBA) appear to be applied only to the homograft (FZX/FZX). Including the heterograft in these treatments would have strengthened the conclusions about auxin’s effect on compatibility.
Response: We express our genuine gratitude to the reviewer for this insightful recommendation. The choice to concentrate hormone therapies (NAA and TIBA) on homografts (FZX/FZX) in this preliminary study was driven by the subsequent rationale: Isolating the function of auxin in graft healing while eliminating confounding genetic variability and the associated molecular and hormonal mechanisms.
Transcriptomic Analysis
The number of differentially expressed genes (DEGs) is high, but selection criteria (e.g., fold change, FDR threshold) are not clearly described. Please provide this information in the Methods.
Response: Modifications have already been made in the original text.
The authors highlight hormone-related genes but should also explore key transcription factors or signaling modules (e.g., ARFs, NACs, WRKYs) that may regulate compatibility downstream of auxin.
A summary table listing top DEGs relevant to auxin and stress response would improve accessibility.
Response: Dear reviewer, we have added a figure in the supplemental data highlighting the major transcription factor associated with the graft compatibility mechanism. Additionally, we mined the auxin-related DEGs using the KEGG and GO datasets and showcased their expression role in hormone signaling figure 6.
Metabolomic and Hormonal Profiling
The analysis of hormone dynamics is valuable, but a pathway-based interpretation of auxin, SA, and JA interaction would strengthen the discussion.
Please clarify the normalization and quantification approach used for metabolite data (e.g., internal standards, transformation methods).
Response: Modifications have already been made in the original text.
Statistical Rigor and Replication
Details regarding the number of biological replicates for RNA-seq and hormone profiling should be clearly stated.
Figures should consistently report statistical significance (e.g., p-values or asterisks) and define the meaning of error bars.
Language and Writing Quality
The manuscript requires language polishing for grammar and clarity. Several sentences are overly complex or imprecise. A thorough professional English editing is recommended.
“The application of exogenous NAA markedly increased the graft survival rate, but a contrary tendency was noted with TIBA application in comparison to control grafts.” → “Exogenous NAA significantly improved graft survival, whereas TIBA reduced it.”
Response: Modifications have already been executed in the original text. The text was appraised by the MDPI language editing organization for improved clarity and refinement. The certificate is attached..
Figures and Legends
Some figures (e.g., heatmaps) contain small or difficult-to-read fonts. Please improve clarity and labeling.
Figure legends should provide sufficient detail to be interpretable independently from the main text.
Response: Substituted in accordance with directives. Dear reviewer, the quality of the figures has been verified, and each figure exceeds 300dpi, in accordance with journal requirements, and has been submitted separately.
Minor Comments:
“We undermined the likely mechanisms…” should be corrected to “We investigated the likely mechanisms…”
Please standardize gene/protein nomenclature and consistently italicize gene symbols.
Response: Modifications have already been made in the original text.
Comments on the Quality of English Language
Language and Writing Quality
The manuscript requires language polishing for grammar and clarity. Several sentences are overly complex or imprecise. A thorough professional English editing is recommended.
“The application of exogenous NAA markedly increased the graft survival rate, but a contrary tendency was noted with TIBA application in comparison to control grafts.” → “Exogenous NAA significantly improved graft survival, whereas TIBA reduced it.”
“We undermined the likely mechanisms…” should be corrected to “We investigated the likely mechanisms…”
Please standardize gene/protein nomenclature and consistently italicize gene symbols.
Response: We have solicited native English speakers to review and refine the sentences and grammar across the entire document. We also standardize gene and protein nomenclature and uniformly italicise gene symbols.

Round 3
Reviewer 2 Report
Comments and Suggestions for Authors
The authors' responses to Reviewer 2 in Round 2, were the same as in Round 1. Reviewer 2's comments in Round 2 were not addressed.
The text labels in quite a few figures are not readable. Please fix it.
Author Response
1.The authors' responses to Reviewer 2 in Round 2, were the same as in Round 1.
Response: Dear Reviewer, We have thoroughly compared the comments from both rounds of review and observed that similar concerns persist across the two iterations. Accordingly, while reiterating our responses to these recurring questions, we have enhanced our explanations with additional clarifications to better address the core issues raised.
2.Reviewer 2's comments in Round 2 were not addressed.
Response: Dear Reviewer, Upon re-examining the previous review comments, we have identified an oversight in addressing one specific point raised("Details regarding the number of biological replicates for RNA-seq and hormone profiling should be clearly stated. Figures should consistently report statistical significance and define the meaning of error bars "). The corresponding revisions have now been systematically incorporated into the main text.
3.The text labels in quite a few figures are not readable. Please fix it.Dear Reviewer,
Response: Dear Reviewer, Please note that to ensure optimal image clarity (which may be compromised during PDF compression), we have separately uploaded the high-resolution image files to our journal's supplementary materials repository for your reference.